# Seed Provision Efficacy of Detached Reproductive Shoots in Restoration Projects for Degraded Eelgrass (*Zostera marina* L.) Meadows

Mingjie Liu [1,2,3,4,5,6], Shaochun Xu [1,2,3,4,5,6], Shidong Yue [1,2,3,4,5,6], Yongliang Qiao [7], Yu Zhang [1,2,3,4,5,6], Xiaomei Zhang [1,2,3,4,5,6] and Yi Zhou [1,2,3,4,5,6,*]

1   CAS Key Laboratory of Marine Ecology and Environmental Sciences, Institute of Oceanology, Chinese Academy of Sciences, Qingdao 266071, China
2   Laboratory for Marine Ecology and Environmental Science, Qingdao National Laboratory for Marine Science and Technology, Qingdao 266237, China
3   Center for Ocean Mega-Science, Chinese Academy of Sciences, Qingdao 266071, China
4   CAS Engineering Laboratory for Marine Ranching, Institute of Oceanology, Chinese Academy of Sciences, Qingdao 266071, China
5   University of Chinese Academy of Sciences, Beijing 100049, China
6   Shandong Province Key Laboratory of Experimental Marine Biology, Qingdao 266071, China
7   School of Environment and Safety Engineering, Qingdao University of Science and Technology, Qingdao 266000, China
*   Correspondence: yizhou@qdio.ac.cn

**Abstract:** Eelgrass (*Zostera marina* L.) is a dominant seagrass species in the temperate waters of the northern hemisphere and is experiencing global declines. The use of eelgrass seeds is increasingly being recognized as a viable option for large-scale restoration projects. Eelgrass reproductive shoots are often collected to obtain seeds or to broadcast seeds in restoration projects. Here, a new method to accurately evaluate the seed provision efficacy of detached eelgrass reproductive is proposed. Viable seeds of detached eelgrass reproductive shoots were collected using in situ net cages at Swan Lake, China. The seed yield and quality of detached eelgrass reproductive shoots under five different treatments (with or without rhizomes and roots, vegetative shoots, and sediment) were compared to select the optimal treatment for this method. The results showed that each detached eelgrass reproductive shoot produced approximately 50 viable seeds on average and the potential seed yield overestimated the actual seed yields by around four times. Seed quality evaluated by size and weight was consistent with that of the natural eelgrass meadow in Swan Lake. Comparing the five different treatments, the simplest treatment (a single reproductive shoot) was convenient and robust for this method. The results indicate that this method is worth further extending to other populations to improve the efficiency of seed use and for effectiveness evaluation in restoration projects.

**Keywords:** ecological restoration; flowering shoot; seagrass; sexual reproduction; Swan Lake

## 1. Introduction

Seagrasses are marine angiosperms that grow in temperate and tropical coastal habitats, and provide important ecosystem services [1]. Seagrass meadows support fisheries productivity [2], significantly reduce bacterial pathogen exposure [3], and contribute to global carbon sequestration [4]. They are estimated to be key global carbon stocks, storing as much as 19.9 Pg of organic carbon [5]. Whilst seagrasses play an important role in regulating climate, they have declined in abundance over recent decades due to global climate change and anthropogenic activities such as coastal development, eutrophication, and fishing activities [6–9]. Seagrass conservation and restoration are essential to help mitigate climate change and maintain ecosystem services [8,10,11]. Restoration efforts have

been performed worldwide to offset or mitigate seagrass habitat losses, and have been shown to enhance the associated ecosystem services [10,12,13].

*Zostera marina* L. (eelgrass), which is capable of both asexual and sexual reproduction, is the most widely distributed seagrass species in the northern hemisphere and the most frequently utilized species in restoration projects [1,10,14–19]. Asexual reproduction via the production of vegetative or clonal shoots is the main contributor to the population recruitment of seagrass meadows [20–22]. Early restoration efforts take advantage of the asexual propagation features of eelgrass and transplant adult plants from healthy meadows to restoration locations in various ways such as the horizontal rhizome method [23], the mechanical transplantation method [24–26], the oyster shells anchoring method [27], and the stone anchoring method [28]. However, these methods are generally labor or cost intensive, and are potentially harmful to the donor meadows [10,28–30].

In addition to asexual reproduction, sexual reproduction, where seeds are produced through reproductive shoots, is also an important population recruitment mechanism for eelgrass [14,16,31–33]. The seeding restoration method involves collecting eelgrass reproductive shoots, and then separating, storing, and sowing the seeds. When the environmental conditions are suitable for seed germination, the seeds are sown in the target restoration area through a variety of methods such as hand-broadcasting [13,29], a mechanical seed planter [34,35], buoy-deployed seeding [29,36,37], burlap bags seeding [38–40], dispenser injection seeding [37], and seed ball burial [41]. The seeding method is convenient, has little impact on natural meadows, and can maintain the genetic diversity of seagrasses [42,43]. Therefore, it has gradually developed into an effective method of restoration, especially for large-scale projects [29,35,37,41,43–46]. One example of successful seagrass restoration involved broadcasting over 70 million eelgrass seeds into mid-Atlantic coastal lagoons in the United States, leading to the recovery of 3612 ha of seagrass [13].

Large-scale restoration programs using seeds will require an assessment of the number of seeds needed for specific planting efforts and an estimate of the number of seeds that are available in a donor meadow [13,43]. However, because the flowering time and seed maturation on reproductive shoots is not synchronized, it is not practical to collect seeds directly in the field [43,47]. Eelgrass reproductive shoots are often collected to obtain seeds [13,37,48] or to broadcast seeds [29,36,49] in restoration projects. Therefore, accurate measurement of the seed yield and quality of the detached eelgrass reproductive shoot is helpful to evaluate the reproductive effort and restoration effect, such as germination rate and seedling establishment rate. The potential seed yield of reproductive shoots, roughly estimated by multiplying the number of spathes per reproductive shoot by the number of seeds/flowers per spathe, might overestimate the actual seed yield [16,50,51]. It is therefore necessary to design a new method to evaluate the seed yield and quality of detached eelgrass reproductive shoots. In addition, when collecting reproductive shoots with a quick upward snapping motion, a small portion of the rhizome and vegetative shoots will detach with the reproductive shoot; however, to the authors' best knowledge, the potential effect of this on the subsequent maturation of the seeds is unknown.

The aim of this study was to propose a method to test the efficacy of detached eelgrass reproductive shoots for seed provision and select the optimal treatment of reproductive shoots for this method. Seeds were collected by temporarily raising detached eelgrass reproductive shoots in net cages in situ, and the quality of the seeds obtained from five different treatments (with or without rhizomes and roots, vegetative shoots, and sediment) was compared. It was hypothesized that a reproductive shoot with rhizomes, roots, vegetative shoot, and sediment would yield the greatest number and quality of seeds because it is closest to the natural state of the reproductive shoots. This study provides important theoretical reference and data support for eelgrass conservation and seeding restoration.

## 2. Materials and Methods

### 2.1. Study Site

The experimental site and reproductive shoot collection location were in Swan Lake (Figure 1A,B; 37°21′0.94″ N, 122°34′43.3″ E), Rongcheng City of Shandong Province, north China. Swan Lake is a marine lagoon with an area of approximately 4.8 km$^2$ and an average water depth <1.5 m dominated by the seagrasses *Z. marina* and *Z. japonica*. The lagoon experiences irregular semidiurnal mixed tides with an average tidal range of around 0.9 m. The substrate of Swan Lake is mainly sandy [52]. There is a large area of eelgrass meadow in Swan Lake, with the maximum distribution area ranging from 199.09 to 231.67 ha in summer [53]. Sexual reproduction plays a relatively important role in the Swan Lake eelgrass population [16]. The reproductive shoots of eelgrass in Swan Lake are first observed in early May, and the reproductive shoot density decreases quickly and disappears completely by the end of July [16].

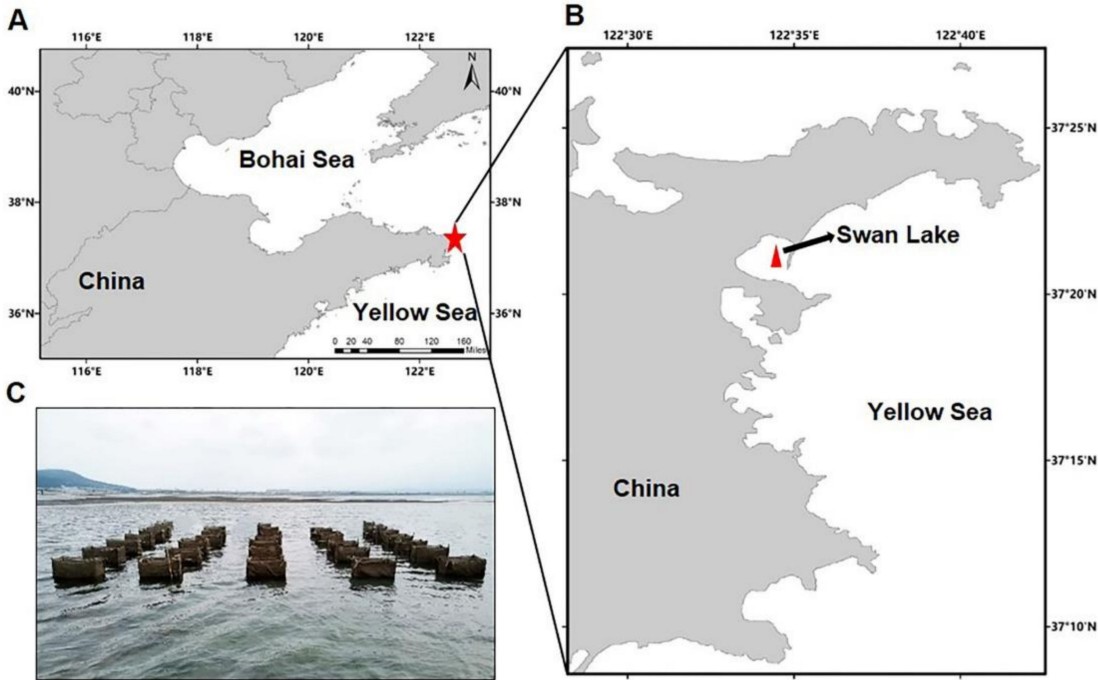

**Figure 1.** The experimental site in Swan Lake (**A**,**B**) and a panoramic view of the experimental installation (**C**).

### 2.2. Experimental Design

A net cage was designed to temporarily maintain detached eelgrass reproductive shoots in situ and collect seeds released from the reproductive shoots. A steel bar with a diameter of 1 cm was welded into a 60-cm × 30-cm × 70-cm (length × width × height) rectangular steel frame. The sides and bottom were wrapped with a 1-mm mesh bag to prevent the seeds from spreading. The outer layer was covered with a 4-mm mesh bag to protect the inner bag and prevent the seeds from spilling out due to the wear of the mesh. The entire net cage (steel frame and two layers of mesh bag) was placed on bare sediment and was secured by ties to two steel bars embedded in the sediment (the two bars were placed diagonally). There were 30 cages in the experiment, arranged in six rows and five columns, with each cage spaced approximately 1 m apart (Figure 1C).

Five treatments were established. Treatment 1 (NNN): a single reproductive shoot without rhizomes and roots. Treatment 2 (RNN): a reproductive shoot connected to a few rhizomes and roots. Treatment 3 (RVN): a reproductive shoot connected to a vegetative shoot by a few rhizomes and roots. On the basis of Treatment 2 and 3, plant units were buried into the sediment as Treatment 4 (RNS) and 5 (RVS), respectively (Figure 2).

On 21 June 2020, eelgrass reproductive shoots were collected randomly in the natural eelgrass meadow in Swan Lake when they had begun to form seeds, but had not yet begun to release them. The collected reproductive shoots were placed in the cages according to the above treatments within 24 h. Each treatment had six replicates and each cage contained 10 reproductive shoots of eelgrass. All of the reproductive shoots were anchored by tying them to a stone with a tie to prevent the reproductive shoots from floating. Prior to the placement of the reproductive shoots, in situ sediment approximately 10 cm deep was added to the cages of Treatment 4 and 5, and the reproductive shoots along with stones and roots were randomly planted into the sediment. In addition, reproductive shoots (n = 20) were randomly selected, and the height of reproductive shoots, the number of spathes per shoot, and the number of seeds per spathe were determined to estimate the potential seed yield per reproductive shoot in the natural eelgrass meadows.

### 2.3. Seed Collection and Evaluation

On 5 August 2020, seeds in all treatments had been fully released from the reproductive shoots and sank to the bottom of the cages. The inner mesh bag was then recovered, and a 2-mm sieve was used to remove large detritus and most of the sediment in the mesh bag. Seeds mixed with small detritus were retained in the 0.7-mm sieve, and then the remaining materials in the sieve were sent to the laboratory for analysis. The seeds were kept in cold storage at 4 °C until required for measurement.

Seeds were recorded as viable if they were firm when gently squeezed with forceps, and intact with a solid embryo inside the seed coat [29,54]. The viable seeds were removed from the materials and counted in each cage. The number of viable seeds was defined as the seed yield of 10 reproductive shoots.

Seed size and weight are the most common seed characteristics and remain relatively constant from year to year [55,56]. Therefore, seed length, diameter, volume, wet weight, dry weight, and moisture content were selected as the indices to evaluate seed quality. Thirty viable seeds were randomly selected from each replicate and spread on soft paper towels to remove water and dry. The wet weight of the seeds was quickly measured with an analytical balance. The seeds were then placed in an oven at 60 °C to dry to a constant weight, and their dry weight was then measured. The mean dry weight, wet weight, and moisture content per seed in each replicate were calculated. The lengths and diameters of 30 viable seeds from each replicate were measured using an industrial digital camera (HTC1000, WeiTu, Shanghai, China; precision of 0.01 mm), and seed volumes were calculated based on the volume of an ellipsoid, $\pi/6 \ LD^2$, where L is the major axes (length) and D is the minor axes (diameter) [57].

### 2.4. Environmental Variables

During the study period, the water temperature (°C) of Swan Lake was measured every 15 min using a HOBO Pendant light/temp MX 2202 (ONSET, Wareham, MA, USA). An ECOPARSB sensor (Sea-Bird Scientific, Bellevue, WA, USA) was used to record light intensity (photosynthetic photon flux density; mol photons $m^{-2} \ d^{-1}$) at the canopy level. In addition, the salinity and pH of seawater were measured every 10 s using an YSI Multi-parameter Water Quality Analyzer (YSI Inc., Yellow Springs, OH, USA) on the night of 4 August 2020.

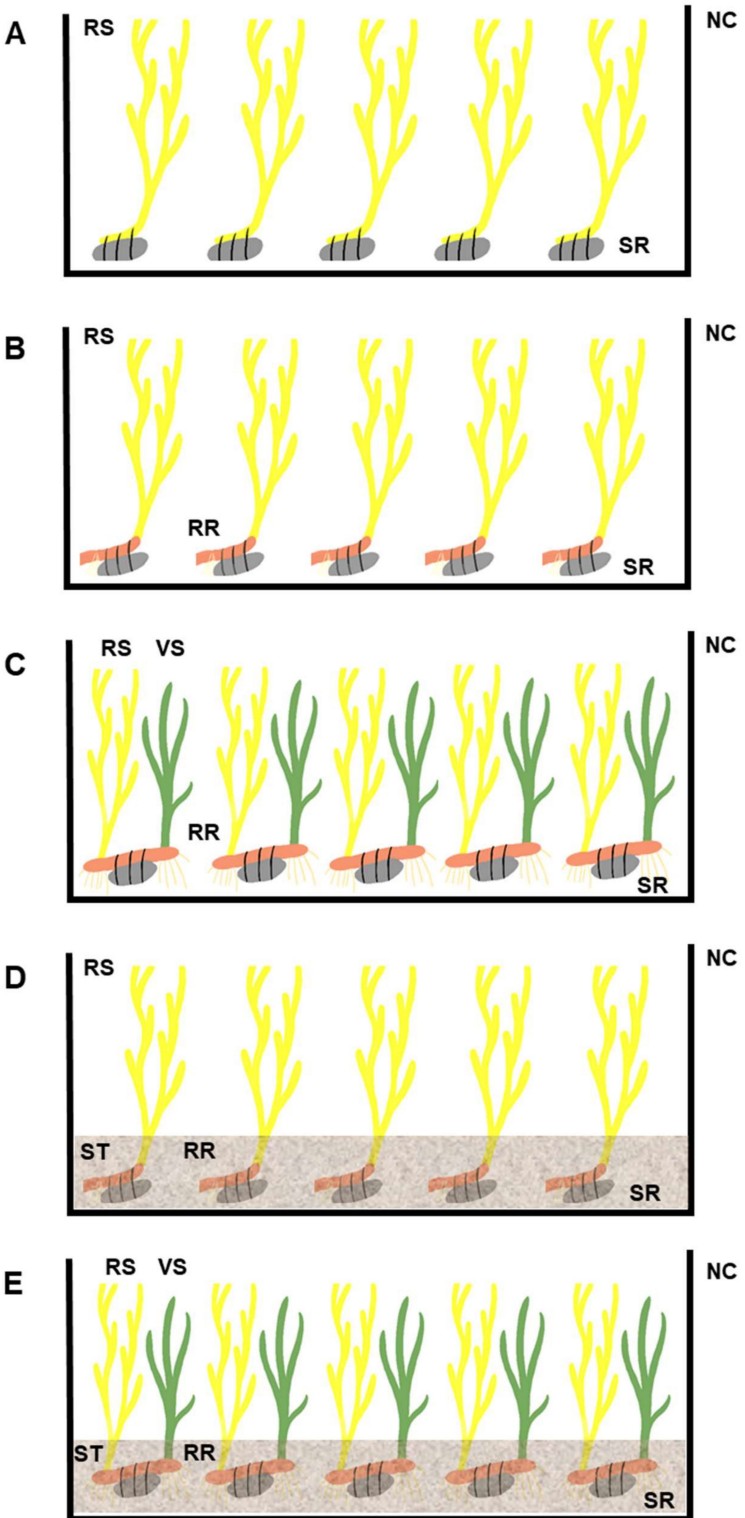

**Figure 2.** Schematic diagram of the five different treatments: (**A**) a single reproductive shoot without rhizomes and roots (NNN); (**B**) a reproductive shoot connected to a few rhizomes and roots (RNN); (**C**) a reproductive shoot connected to a vegetative shoot by a few rhizomes and roots (RVN); (**D**) a reproductive shoot with rhizomes, roots, and sediment (RNS); (**E**) a reproductive shoot with rhizomes, roots, vegetative shoot, and sediment (RVS); (RS) reproductive shoot; (VS) vegetative shoot; (RR) rhizome and root; (SR) stone and rope; (ST) sediment; and (NC) net cage.

*2.5. Data Analysis*

Data were represented by mean ± standard deviation. The height of eelgrass vegetative shoots in RVN and RVS were compared by an independent sample *t*-test. Seed yield, seed weight, and seed size were compared between the different treatments by one-way analysis of variance, with seed size being the mean length and diameter of 30 viable seeds per replicate. The normality and homogeneity of variance were tested using Shapiro–Wilk and Levene's tests, respectively. When the data did not meet the demands of normality and homogeneity, a Kruskal–Wallis test was used, and post hoc tests were performed using Duncan's test to compare the differences between treatments when the demands were met. R 4.0.3 was used for data analyses. Differences were considered significant at a probability level of $p < 0.05$.

## 3. Results

According to observations, the seeds of each treatment had not fully matured and fallen off by 21 July 2020 (30 days after deployment). In particular, in the treatment with vegetative shoots, the seed maturity was lower. On 5 August 2020 (45 days after deployment), the seeds of all treatments had matured and fallen off. This was consistent with the development of reproductive shoots in the natural eelgrass meadow in Swan Lake [16]. It was also observed that some vegetative shoots in RVN and RVS were still green and fresh, and new vegetative shoots were cloned. The heights of surviving vegetative shoots were compared between the two treatments; there was no significant difference ($t = -1.602$, $df = 65$, $p = 0.114$).

*3.1. Environment Variables*

Over the course of the experiment, seawater temperature ranged from 18.19 °C to 32.39 °C, with an average of 22.58 ± 2.10 °C. The average photosynthetic photon flux density was 98.57 ± 24.24 mol photons $m^{-2}$ $d^{-1}$. Seawater salinity was 28.41 PSU and pH was 8.30 on the night of 4 August 2020.

*3.2. Seed Yield*

The height of eelgrass reproductive shoots and the potential seed yield per reproductive shoot (the number of spathes per shoot × the number of seeds per spathe) in natural seagrass meadows before the experiment are shown in Table 1. The mean height of reproductive shoots was 71.70 ± 12.57 cm, and each reproductive shoot could potentially produce 204.73 ± 11.17 seeds on average.

**Table 1.** Reproductive shoot traits in a natural eelgrass meadow in Swan Lake, June 2020 (n = 20).

| | Reproductive Shoot Height (cm) | No. Spathes per Reproductive Shoot | No. Seeds per Spathe | No. Seeds per Reproductive Shoot |
|---|---|---|---|---|
| Mean ± SD | 71.70 ± 12.57 | 23.40 ± 7.12 | 8.74 ± 1.57 | 204.73 ± 11.17 |

As shown in Figure 3, there was no significant difference in the number of viable seeds produced by 10 reproductive shoots among the treatments (Kruskal–Wallis chi-squared test = 8.7742, *df* = 4, *p* = 0.067). RVN tended to have the highest seed yield (668.17 ± 345.03), followed by NNN (617.50 ± 187.83), while RVS tended to have the lowest seed yield (334.33 ± 82.02). The mean number of viable seeds produced per reproductive shoot among the five treatments was 50 (33–67), accounting for approximately 24.39% (16.10–32.68%) of the potential seed yield (205 seeds per reproductive shoot).

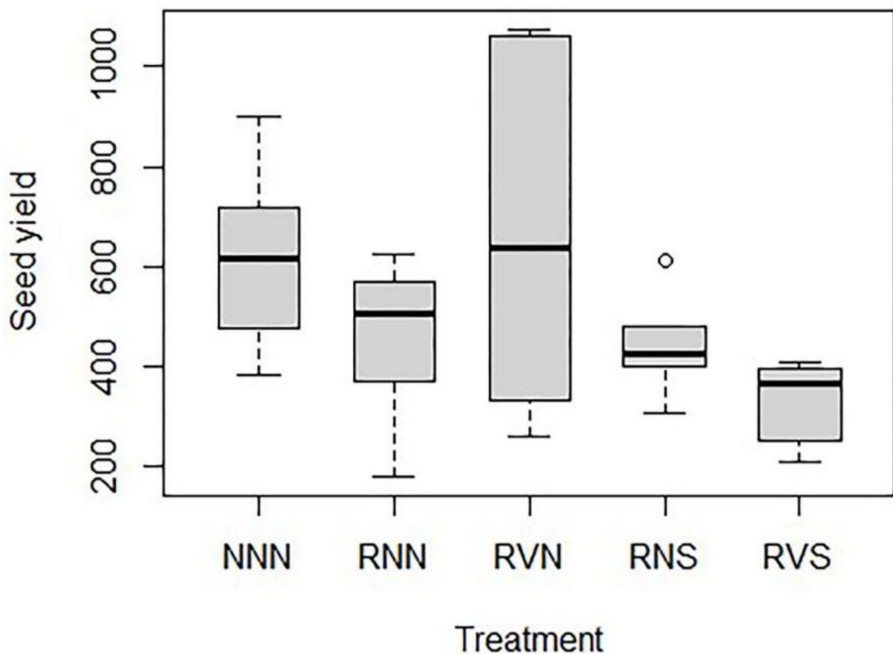

**Figure 3.** Viable seed yield of the different treatments (NNN: Treatment 1; RNN: Treatment 2; RVN: Treatment 3; RNS: Treatment 4; RVS: Treatment 5). The black lines indicate the median of seed yield and the circle indicates an outlier.

### 3.3. Seed Weight

There were no significant differences in mean dry weight, wet weight, and moisture content per seed among the treatments ($F_{4, 25} = 0.568$, $p = 0.688$; $F_{4, 25} = 0.304$, $p = 0.873$; and $F_{4, 25} = 1.325$, $p = 0.288$, respectively). The mean dry weight per seed was 3.11 (2.93–3.25) mg and the mean wet weight per seed was 5.59 (5.46–5.75) mg among the treatments. The highest seed moisture content tended to be in NNN (46.40 ± 3.31%), while the lowest seed moisture content tended to be in RNN (41.87 ± 1.20%, Figure 4).

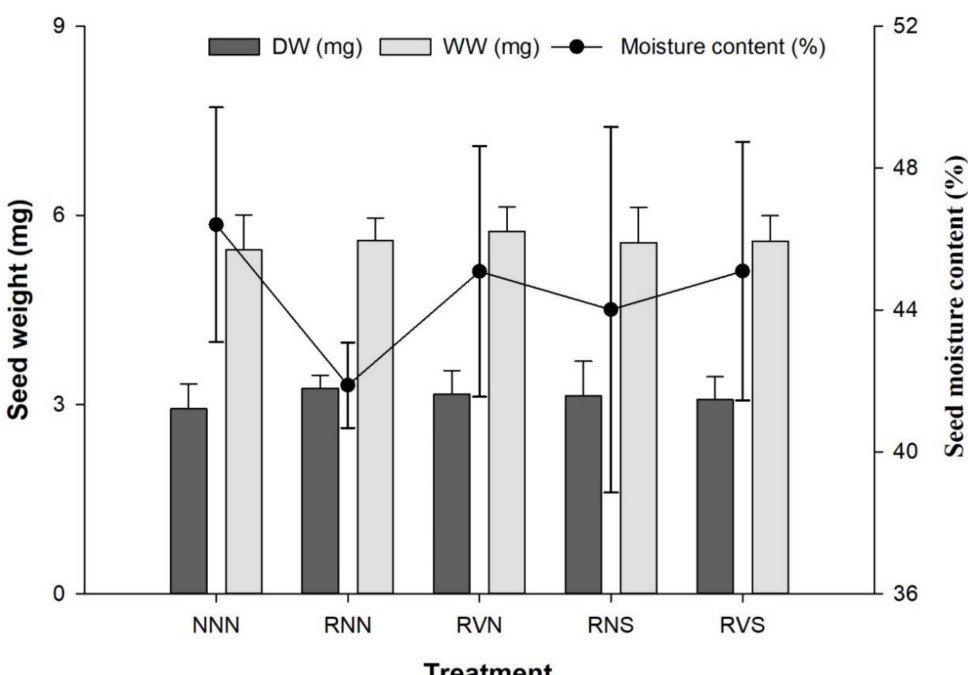

**Figure 4.** Mean seed dry weight (DW), wet weight (WW), and moisture content in the different treatments.

*3.4. Seed Size*

There were no significant differences in mean seed length, diameter, and volume in the different treatments ($F_{4, 25} = 0.937$, $p = 0.459$; $F_{4, 25} = 0.607$, $p = 0.661$; and $F_{4, 25} = 0.598$, $p = 0.667$, respectively). The mean seed length was 3.34 (3.25–3.39) mm and the mean seed diameter was 1.65 (1.62–1.67) mm among the treatments. The maximum seed volume tended to be in RVN ($5.00 \pm 0.14$ mm$^3$) and the minimum seed volume tended to be in NNN ($4.55 \pm 1.00$ mm$^3$, Figure 5).

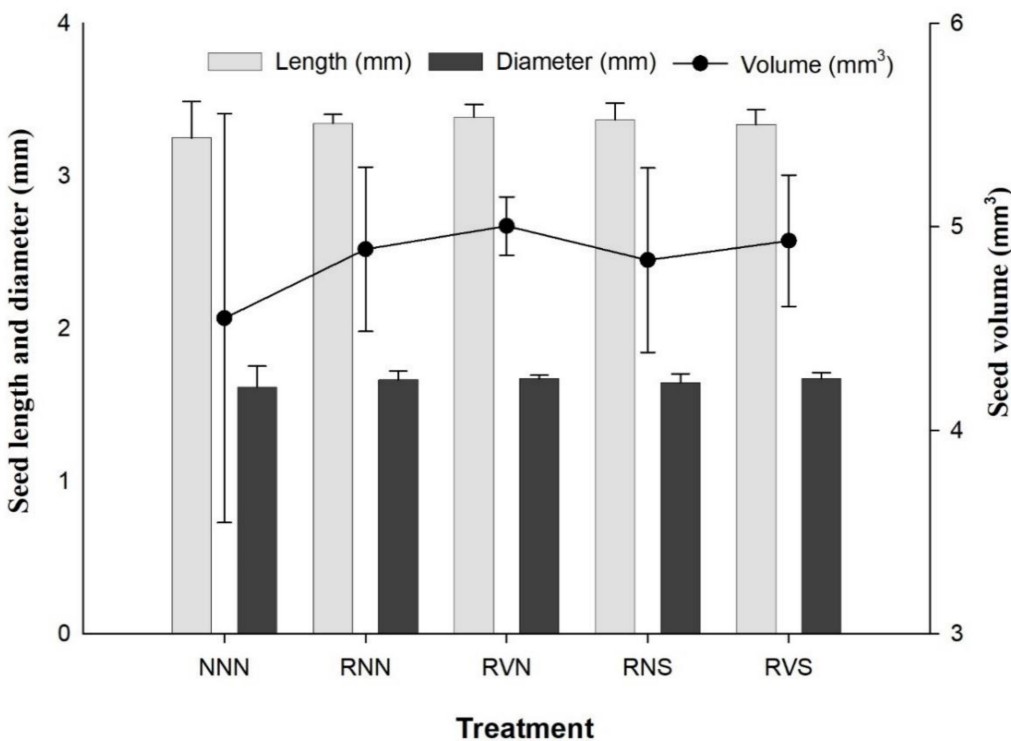

**Figure 5.** Mean seed length, diameter, and volume in the different treatments.

## 4. Discussion

Sexual recruitment plays an important role in population expansion and recovery, highlighting the importance of properly assessing the seed yield of seagrass [58–60]. Eelgrass seeds do not ripen simultaneously, even on the same reproductive shoot [43,47]; therefore, it is difficult to accurately estimate the actual seed yield per reproductive shoot. If the reproductive shoots are collected too early, the seeds will be immature, and when collected too late, the seeds will be released [61].

Wyllie-Echeverria et al. (2003) and Infantes et al. (2018) collected mature seeds by bagging reproductive shoots separately in mesh bags and sieving the contents of the bag after seed maturation, but this method may hinder the development of reproductive shoots and have an adverse effect on seed formation [55,61]. In this study, a new method to evaluate the seed provision efficacy of detached eelgrass reproductive shoots was proposed that simulated the natural position and movements of reproductive shoots in the water column through in situ incubation in net cages. The results showed that the actual seed yield per reproductive shoot was 50 (33–67) in Swan Lake (Figure 3). In addition, the results indicated that the potential seed yield overestimated the actual seed yield by approximately four times (Figure 3 and Table 1). These results provide valuable data to estimate the number of reproductive shoots needed to obtain a specific quantity of viable seeds for restoration, and to improve the efficiency of reproductive shoot and seed use.

In this experiment, the detached eelgrass reproductive shoots under the different treatments showed consistent seed provision efficacy and there was no significant effect of treatment on seed formation. Rhizomes and roots seemed to serve only as anchors

and supports, and had no significant effect on the seed formation [15,62]. Sediment may interfere with the evaluation of the results. The seed yield of the RNS and RVS groups tended to decrease compared with the other three treatments, which may be due to the influence of microorganisms in the sediment causing seed inactivation [63]. Vegetative shoots did not affect the quantity and quality of seeds, but a lower seed maturity in the treatments with vegetative shoots was observed on 21 July 2020. The effect of cloning lateral shoots on the seed formation of reproductive shoots is worthy of further investigation. Considering the undifferentiated results, the simplest treatment (Treatment 1, a single reproductive shoot) can be selected for this method, which is not only simple and feasible, but can also obtain equivalent results.

Seed size and weight are important indices for evaluating seed quality, which not only affect seed dispersal and seedling establishment [57,64], but also may reflect the adaptation mechanisms of eelgrass populations to different environments [16,64]. Heavier and larger seeds contain greater storage reserves and produce seedlings with significantly higher biomass, whereas smaller and lighter seeds increase the opportunity to colonize new areas due to higher dispersal potential [57,64]. Differences in eelgrass seed size among populations have been previously reported [55,61]. Xu et al. (2018) documented that eelgrass seeds from Swan Lake were notably smaller in width and length, and lighter in weight compared with those from Huiquan Bay [16]. The seed size and weight obtained in the present study were consistent with previous studies in Swan Lake [16,65]. This indicates that seed size and weight remain relatively constant from year to year, and the detached eelgrass reproductive shoots obtained seeds of the same quality as the natural meadow using this method. This is probably because it can ensure the continued development of reproductive shoots after collection and 10 reproductive shoots in cage can maintain pollination.

Although the seed quality results were consistent with previous studies, the potential seed yield per reproductive shoot in this study (Table 1) was higher than that in the previous studies in Swan Lake [16,66]. This difference was mainly due to the number of spathes per reproductive shoot rather than the number of seeds per spathe. This may be due to differences in the water depth where the reproductive shoots were collected from [61,67,68] or the interannual variation in the number of spathes per reproductive shoot [60]. The reproductive shoots are usually higher and have more spathes in deep water than in shallow water eelgrass meadows [61,69]. There is also large variation in the potential seed yield between different geographic populations [16,22,32,67,70]. For example, the number of seeds per reproductive shoot was 5–98 in Sweden [61], 19–41 in Denmark [67], and 20–100 in Chesapeake Bay [29]. Therefore, it is suggested that this method can be integrated into restoration projects and extended to other areas to assess the capacity of reproductive shoot seed supply.

The transplanting of eelgrass vegetative shoots by stone anchoring has had high success with a transplant survival rate of >95% [28]. Based on the above research, an alternative method for eelgrass restoration is proposed that can achieve natural dispersal of eelgrass seeds and is similar to the stone anchoring method (the stone-anchored seeding method; Figure 6). Eelgrass reproductive shoots are collected and transferred to the target restoration location where they are anchored underwater by stones so that the shoots are suspended on the surface of the sediment. When the seeds are ripe, they naturally fall to the seafloor. Multiple reproductive shoots can be bound to a stone, and the number of planting units per unit area can be flexibly adjusted according to the expected planting density.

The buoy-deployed seeding method taking advantage of the reproductive shoots of eelgrass to disperse seeds has been developed and applied in restoration [29,36,37]. In the buoy-deployed seeding method, eelgrass reproductive shoots are directly placed into mesh bags suspended at the restoration location after collection. As the seeds mature, they dehisce naturally and fall to the seafloor due to negative buoyancy where they then germinate and form seedlings when the environmental conditions are suitable. The similarity between the buoy-deployed seeding method and the stone-anchored seeding method proposed here is that the reproductive shoots are deployed immediately after collection, and the

reproductive shoots and seeds do not need to be preserved. The buoy-deployed seeding method, in which the reproductive shoots are densely packed in nets, may be harmful to the development of seeds [29], while the proposed method better mimics the natural position and movements of reproductive shoots in the water column.

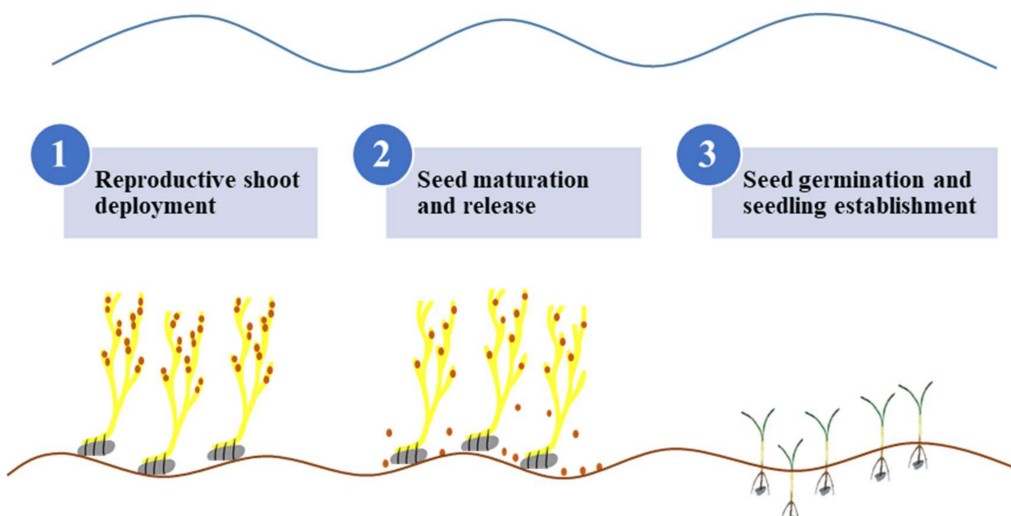

**Figure 6.** Schematic diagram of the proposed stone-anchored seeding method using detached eelgrass reproductive shoots. The blue wavy line indicates sea level and the brown wavy line indicates sediment.

Based on the experience of the buoy-deployed seeding method, the low seedling recruitment rate might be a potential problem in the proposed stone-anchored seeding method; therefore, the seed germination rate and seedling establishment rate need to be tested. The timing of reproductive shoot collection is also critical to ensure the greatest seed yield. The proposed stone-anchored seeding method is not only environmentally friendly, but also eliminates the infrastructure requirements for the long-term storage of reproductive shoots and seeds. However, further field experiments are required to verify the effectiveness of this method.

## 5. Conclusions

In the present study, a new method for evaluating the seed provision efficacy of detached eelgrass reproductive shoots was proposed and an optimal treatment for this method was selected. This method can improve the efficiency of the use of reproductive shoots and seeds in restoration programs, and is worthy to further extend to other populations. Additionally, an alternative seeding method (the stone-anchored seeding method) for eelgrass restoration that is environmentally friendly and circumvents the infrastructure requirements for processing and holding large numbers of seeds, was conceptually presented. It is recommended to further trial this stone-anchored seeding method using detached reproductive shoots in eelgrass restoration projects.

**Author Contributions:** Methodology, Y.Z. (Yi Zhou), M.L. and S.X.; investigation, M.L., S.X., S.Y., Y.Q., Y.Z. (Yu Zhang) and X.Z.; writing—original draft, M.L.; writing—review and editing, M.L., Y.Z. (Yi Zhou), S.X. and S.Y.; funding acquisition, Y.Z. (Yi Zhou). All authors have read and agreed to the published version of the manuscript.

**Funding:** This research was supported by the National Key R&D Program of China (2022YFC3105403), the National Natural Science Foundation of China (32270405/42206142/32000269), the Natural Science Foundation of Shandong Province (ZR2020QD106), the China Postdoctoral Science Foundation (2022M723183), the National Science and Technology Basic Work Program (2015FY110600), and the Taishan Scholars Program (Distinguished Taishan Scholars).

**Institutional Review Board Statement:** Not applicable.

**Informed Consent Statement:** Not applicable.

**Data Availability Statement:** The data presented in this study are available upon request from the corresponding author. The data are not publicly available because of privacy restrictions.

**Conflicts of Interest:** The authors declare no conflict of interest.

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
