# Peer review of "Seed Provision Efficacy of Detached Reproductive Shoots in Restoration Projects for Degraded Eelgrass (Zostera marina L.) Meadows"

_sustainability, doi:10.3390/su15075904_

Round 1

Reviewer 1 Report

This manuscript deals with the effectiveness of detached reproductive shoots in the context of the restoration of Zostera marina (L.) meadows. I find the manuscript well organised and written and I believe that the method proposed by the authors is innovative. I have only a few issues: 

- delete the description of the treatments in the Introduction; 

- explain and describe the treatments also in the caption of Figure 2 ; 

- clarify the concept of potential seed yield in the results. 

I believe that the manuscript deserves the publication after minor revision. 

Author Response

We are grateful to the reviewer's positive and constructive comments. We have revised our manuscript accordingly. Detailed responses to the mentioned comments and suggestions are listed below point by point.

Delete the description of the treatments in the Introduction.

Response: We have made the changes as suggested.

Explain and describe the treatments also in the caption of Figure 2.

Response: We have made the changes as suggested. (Lines 142-146)

Clarify the concept of potential seed yield in the results.

Response: We have made the changes as suggested. (Line 209)

Reviewer 2 Report

General summary of manuscript:

The manuscript “Efficacy of detached reproductive shoots for provisioning seeds in restoration project for degraded eelgrass (Zostera marina L.) beds” by Liu et al compared a few methods and proposed a new method to evaluate the efficacy of detached eelgrass reproductive shoots for provisioning seeds. This study would provide significant insights and practical knowledge on how to restore eelgrass. However, it still needs careful revision before it can become suitable for publication. There are some issues with the current manuscript that need to be addressed.

Specific comments:

Line 102: Please mention the exact figure number here. For example figure 1A , 1B....

Line 108-109: Needs a citation.

Line 125: Same as line 102.

Figure 2: I don't understand why the authors only mention SR (stone and rope) in figure B (treatment 2). Is it because they only use stone and rope in treatment 2? Same comment for ST in figure D.

Line 151: "remove large detritus and most of the sediment" from where?

Line 167: “digital camera”  Can the authors provide the model, manufacturer name, etc.?

Line 210-212: Which data? Is there any table/figure...?

Line 231-235: "3.4. Environment variables" this part should be written first in the results section.

Line 232-235: Which data? Is there any table/figure...?

Line 254-256:Please mention the figures/table here.

Line 264-266: Please change “which may be due to the action of microorganisms in the sediment causing seed inactivation” to “ which may be due to the influence of microorganisms in the sediment causing seed inactivation”.

Line 266-268: Citation?

Line 282-263:Please change “The seed size and weight obtained in the present study were consistent with the historical results in Swan Lake” to “ The seed size and weight obtained in the present study were consistent with the previous studies in Swan Lake”.

Line 286-288: This is not clear to me.

Line 289-291: The sentence does not flow well. Please rewrite the sentence. Also please modify the term “historical results” .

Line 312-313: “In this method, eelgrass reproductive shoots were directly placed into mesh bags suspended at the restoration location after collecting.” Which methods?  Needs a citation.

Figure 6: The author mentioned in the abstract that “a single reproductive shoot” or “treatment 1” is the effective method. However, when we look at figure 6, it seems like the authors are suggesting  “treatment 2” is the effective method. Can the authors clarify these points?

Author Response

We are grateful to the reviewer's positive and constructive comments. We have revised our manuscript accordingly. Detailed responses to the mentioned comments and suggestions are listed below point by point.

Delete the description of the treatments in the Introduction.

Response : We have made the changes as suggested.

Explain and describe the treatments also in the caption of Figure 2.

Response: We have made the changes as suggested. (Lines 142-146)

Clarify the concept of potential seed yield in the results.

Response: We have made the changes as suggested. (Line 209)

Line 102: Please mention the exact figure number here. For example figure 1A , 1B....

Response: We have made the changes as suggested. (Line 99)

Line 108-109: Needs a citation.

Response : We have made the revision. (Line 106)

Line 125: Same as line 102.

 Response: We have made the changes as suggested. (Line 122)

Figure 2: I don't understand why the authors only mention SR (stone and rope) in figure B (treatment 2). Is it because they only use stone and rope in treatment 2? Same comment for ST in figure D.

Response: We have added the short name of each component in all figures (Figure 2A-E).

Line 151: "remove large detritus and most of the sediment" from where?

Response: We have corrected the inappropriate expression. (Lines 150-151)

Line 167: “digital camera”  Can the authors provide the model, manufacturer name, etc.?

Response: We have provided more details about the camera. (Line 168)

Line 210-212: Which data? Is there any table/figure...?

Response: The Figure 3 shows the number of viable seeds produced by ten reproductive shoots in different treatments. The mean number of viable seeds produced per reproductive shoot is obtained by dividing the data in Figure 3 by 10. The ratio is further divided by the potential seed yield. In order to prevent misunderstanding, we have modified accordingly. (Line 214)

Line 231-235: "3.4. Environment variables" this part should be written first in the results section.

Response: We have made the changes as suggested.

Line 232-235: Which data? Is there any table/figure...?

Response: The central idea of our study is to confirm the feasibility of the assessment method, so we briefly provided the environment background values without plotting them.

Line 254-256:Please mention the figures/table here.

Response: We have made the changes as suggested. (Lines 257, 259)

Line 264-266: Please change “which may be due to the action of microorganisms in the sediment causing seed inactivation” to “ which may be due to the influence of microorganisms in the sediment causing seed inactivation”.

Response: We have made the changes as suggested. (Line 267)

Line 266-268: Citation?

Response: We have corrected the inappropriate expression. (Lines 268-271)

Line 282-263:Please change “The seed size and weight obtained in the present study were consistent with the historical results in Swan Lake” to “ The seed size and weight obtained in the present study were consistent with the previous studies in Swan Lake”.

Response: We have made the changes as suggested. (Lines 283-284)

Line 286-288: This is not clear to me.

Response: According to our observations on July 21, 2020 (30 days after deployment), the seeds of each treatment were not fully ripe and some spathes on reproductive shoots were blooming. Meanwhile, on August 5, 2020 (45 days after de-ployment), we also found that some vegetative shoots in the RVN and RVS were still green and fresh. Therefore, we believe that the reproductive shoots continued to develop after collection and deployment and produced seeds of the same quality as the natural meadow.

Line 289-291: The sentence does not flow well. Please rewrite the sentence. Also please modify the term “historical results” .

Response: We have made the revision. (Lines 290-292)

Line 312-313: “In this method, eelgrass reproductive shoots were directly placed into mesh bags suspended at the restoration location after collecting.” Which methods?  Needs a citation.

Response: We have made the revision. (Lines 313-314)

Figure 6: The author mentioned in the abstract that “a single reproductive shoot” or “treatment 1” is the effective method. However, when we look at figure 6, it seems like the authors are suggesting  “treatment 2” is the effective method. Can the authors clarify these points?

Response: We have modified Figure 6 to keep the conclusions consistent with the abstract. (Line 331)

Reviewer 3 Report

Reviewer's comments

1.      Some Grammatical problems have been noticed from the abstract itself. Try to fix them.

2.      3. If you have mentioned the complete binomial name of a species once, try to avoid repeating the full name. (Ref L 47, L 104)

Abstract

3.      L 18- The sentence is incomplete, please paraphrase it.

4.      L 19-20 – The sentence should be paraphrased to make more sense.

5.      L 20-21- “Propose” instead of “Proposed”.

6.      L 22-25- Should be converted in to two sentences.

Introduction

7.      L 37- Use “Pathogen” instead of “Pathogens”.

8.      L 36-39- Should be converted into two sentences. The second sentence can be paraphrased like this: “They are estimated to be key global carbon stocks, storing as much as 19.9 Pg of organic carbon.”

9.      L 40- “Seagrass meadows” rather than “seagrasses meadows”

10.   L 42-43 – Grammatical error: paraphrase like this: “Seagrass conservation and restoration are essential to help mitigate climate change and maintain ecosystem services”.

11.   L 51-56- To make it clearer, rephrase it and split it into two sentences.

12.   L 58- Comma missing after “Eelgrass”.

13.   L 70-72- Put the article "an" before the terms "assessment" and "estimate" in the sentence. Change "estimate the number" to "estimate of the number".

14.   L 75-78- Put a "comma" before "such as," and make sure to check it throughout the draft.

15.   L 78-80- Put a "comma" after "shoots" and "spathe," and delete "lead to" from the sentence.

16.   L 84- Use a "semi-colon" instead of a "comma" before "however."

17.   2. Some lines in the introduction seem better to be added under methodology part. L 91-98

18.   L 96- Replace “give” with “because”.

Materials & methods

19.   L 108- Add “the” before “eelgrass” in the sentence.

20.   In figure 2, the labelling seems confusing, you can either add the short name of each component in all figures or add what each pictorial component represented implicates in a separate column.

21.   L 159-161- Rephrase the sentence to make it more aesthetically pleasing.

Discussion

22.   L 244- Put a "comma" after "simultaneously" in the sentence.

23.   L 245-247- Make the sentence more comprehensible by rephrasing it.

24.   L 249- Replace “shoot” with “shoots”

25.   L 251- Use “have an” instead of “make” in the sentence to make it grammatically correct.

26.    L 267- Use “they may” rather than “it may” in this sentence to match pronouns.

27.   L 268-269- To avoid grammatical errors, rephrase the sentence as follows: "That may be the reason why we observed a lower seed maturity in the treatments with vegetative shoots on July 21, 2020."

28.   L 273- The phrase " which is not only simple and feasible, but also can obtain equivalent result” should be changed to “which is not only simple and feasible but can also obtain an equivalent result” in this context.

29.   L 274- Use “indices” rather than “index”.

30.   L 275- Replace the term “seedlings” with seedling.

31.   L 276- Write “mechanisms” instead of “mechanism.”

32.   L 277- Write “seedlings with” instead of “seedlings of”

33.   L 324- Use “remain to be tested” instead of “remains to be test”

34.   L 325-327- Eliminate grammatical errors.

Conclusion

35.   L 334- Replace “chosen” with “chose”

36.   L 338- Change “that” to “which” and replace “circumvent” with “circumvents”

37.   In reference portion  the name of some journals are given in  abbreviated form and some are in full title. It would be better if it follows a uniformity.

Author Response

We are grateful to the reviewer's positive and constructive comments. We have revised our manuscript accordingly. Detailed responses to the mentioned comments and suggestions are listed below point by point.

Point 1: Some Grammatical problems have been noticed from the abstract itself. Try to fix them.

Response: We have made the revision. (Lines 18-32)

Point 2: If you have mentioned the complete binomial name of a species once, try to avoid repeating the full name. (Ref L 47, L 104)

Response: We have corrected this mistake. (Line 101)

Abstract

Point 3: L 18- The sentence is incomplete, please paraphrase it.

Response: We have made the revision. (Lines 18-19)

Point 4: L 19-20 – The sentence should be paraphrased to make more sense.

Response: We have made the revision. (Lines 19-21)

Point 5: L 20-21- “Propose” instead of “Proposed”.

Response: We have corrected this mistake. (Line 21)

Point 6: L 22-25- Should be converted in to two sentences.

Response: We have made the revision. (Lines 23-26)

Introduction

Point 7: L 37- Use “Pathogen” instead of “Pathogens”.

Response: We have corrected this mistake. (Line 38)

Point 8: L 36-39- Should be converted into two sentences. The second sentence can be paraphrased like this: “They are estimated to be key global carbon stocks, storing as much as 19.9 Pg of organic carbon.”

Response: We have made the revision. (Lines 39-40)

Point 9: L 40- “Seagrass meadows” rather than “seagrasses meadows”

Response: We have corrected this mistake. (Line 41)

Point 10: L 42-43 – Grammatical error: paraphrase like this: “Seagrass conservation and restoration are essential to help mitigate climate change and maintain ecosystem services”.

Response : We have corrected this mistake. (Lines 43-44)

Point 11: L 51-56- To make it clearer, rephrase it and split it into two sentences.

Response : We have made the revision. (Lines 51-56)

Point 12: L 58- Comma missing after “Eelgrass”.

Response : We have corrected this mistake. (Line 58)

Point 13: L 70-72- Put the article "an" before the terms "assessment" and "estimate" in the sentence. Change "estimate the number" to "estimate of the number".

Response : We have corrected this mistake. (Lines 70-72)

Point 14: L 75-78- Put a "comma" before "such as," and make sure to check it throughout the draft.

Response : We have made the revision throughout the manuscript.

Point 15: L 78-80- Put a "comma" after "shoots" and "spathe," and delete "lead to" from the sentence.

Response : We have made the revision. (Lines 78-80)

Point 16: L 84- Use a "semi-colon" instead of a "comma" before "however."

Response : We have corrected this mistake. (Line 84)

Point 17: 2. Some lines in the introduction seem better to be added under methodology part. L 91-98

Response : We have made the revision.

Point 18: L 96- Replace “give” with “because”.

Response : We have made the revision.

Materials & methods

Point 19: L 108- Add “the” before “eelgrass” in the sentence.

Response : We have corrected this mistake. (Line 106)

Point 20: In figure 2, the labelling seems confusing, you can either add the short name of each component in all figures or add what each pictorial component represented implicates in a separate column.

Response : We have added the short name of each component in all figures (Figure 2A-E).

Point 21: L 159-161- Rephrase the sentence to make it more aesthetically pleasing.

Response : We have made the revision. (Lines 160-161)

Discussion

Point 22: L 244- Put a "comma" after "simultaneously" in the sentence.

Response : We have made the revision. (Line 246)

Point 23: L 245-247- Make the sentence more comprehensible by rephrasing it.

Response : We have made the revision. (Lines 247-249)

Point 24: L 249- Replace “shoot” with “shoots”

Response : We have corrected this mistake. (Line 251)

Point 25: L 251- Use “have an” instead of “make” in the sentence to make it grammatically correct.

Response : We have corrected this mistake. (Line 253)

Point 26: L 267- Use “they may” rather than “it may” in this sentence to match pronouns.

Response: We have corrected the inappropriate expression. (Lines 268-271)

Point 27: L 268-269- To avoid grammatical errors, rephrase the sentence as follows: "That may be the reason why we observed a lower seed maturity in the treatments with vegetative shoots on July 21, 2020."

Response : We have corrected the inappropriate expression. (Lines 268-271)

Point 28: L 273- The phrase " which is not only simple and feasible, but also can obtain equivalent result” should be changed to “which is not only simple and feasible but can also obtain an equivalent result” in this context.

Response : We have corrected this mistake. (Lines 273-274)

Point 29: L 274- Use “indices” rather than “index”.

Response: We have corrected this mistake. (Line 275)

Point 30: L 275- Replace the term “seedlings” with seedling.

Response : We have corrected this mistake. (Line 276)

Point 31: L 276- Write “mechanisms” instead of “mechanism.”

Response : We have corrected this mistake. (Line 277)

Point 32: L 277- Write “seedlings with” instead of “seedlings of”

Response : We have corrected this mistake. (Line 278)

Point 33: L 324- Use “remain to be tested” instead of “remains to be test”

Response : We have corrected this mistake. (Line 326)

Point 34: L 325-327- Eliminate grammatical errors.

Response : We have corrected this mistake. (Lines 327-329)

Conclusion

Point 35: L 334- Replace “chosen” with “chose”

Response : We have corrected this mistake. (Line 336)

Point 36: L 338- Change “that” to “which” and replace “circumvent” with “circumvents”

Response : We have corrected this mistake. (Line 340)

Point 37: In reference portion the name of some journals are given in abbreviated form and some are in full title. It would be better if it follows a uniformity.

Response : We have made the changes as suggested.